# Effect of Different Dietary Iron Contents on Liver Transcriptome Characteristics in Wujin Pigs

**DOI:** 10.3390/ani14162399

**Published:** 2024-08-19

**Authors:** Lin Gao, Xiaokun Xing, Rongfu Guo, Qihua Li, Yan Xu, Hongbin Pan, Peng Ji, Ping Wang, Chuntang Yu, Jintao Li, Qingcong An

**Affiliations:** 1Yunnan Provincial Key Laboratory of Tropical and Subtropical Animal Viral Diseases, Yunnan Academy of Animal Husbandry and Veterinary Sciences, Kunming 650201, China; mmnwang@163.com; 2Yunnan Provincial Key Laboratory of Animal Nutrition and Feed Science, Faculty of Animal Science and Technology, Yunnan Agricultural University, Kunming 650201, China; xingxk2022@163.com (X.X.); rongfug@163.com (R.G.); kmliqihua05@sohu.com (Q.L.); ynsdyz@163.com (H.P.); sky_xiang@163.com (P.J.); pingwangna@126.com (P.W.); 13312572758@163.com (C.Y.); 15924904191@163.com (J.L.); 3Yunnan East Hunter Agriculture and Forestry Development Co., Ltd., Shuifu 657803, China; taipingxuyan@163.com

**Keywords:** iron, Wujin piglet, serum iron, transcriptome analysis, iron regulation signaling pathways

## Abstract

**Simple Summary:**

Studies on Wujin pigs mainly focus on epigenetic phenotypes, fat metabolism, antioxidant capacity, and immune capacity and rarely report on iron homeostasis regulation. In this study, the effects of different dietary iron levels on the serum iron metabolism parameters of Wujin pigs were assessed. The effects of the different diets on gene expression levels in the liver were assessed to identify the key regulatory pathways of iron metabolism, provide a theoretical basis for nutritional research and resource utilization, and lay a theoretical foundation for exploring the regulatory pathways and characteristics of iron metabolism in Wujin pigs, which have important application value.

**Abstract:**

Iron is an important trace element that affects the growth and development of animals and regulates oxygen transport, hematopoiesis, and hypoxia adaptations. Wujin pig has unique hypoxic adaptability and iron homeostasis; however, the specific regulatory mechanisms have rarely been reported. This study randomly divided 18 healthy Wujin piglets into three groups: the control group, supplemented with 100 mg/kg iron (as iron glycinate); the low-iron group, no iron supplementation; and the high-iron group, supplemented with 200 mg/kg iron (as iron glycinate). The pre-feeding period was 5 days, and the formal period was 30 days. Serum was collected from empty stomachs before slaughter and at slaughter to detect changes in the serum iron metabolism parameters. Gene expression in the liver was analyzed via transcriptome analysis to determine the effects of low- and high-iron diets on transcriptome levels. Correlation analysis was performed for apparent serum parameters, and transcriptome sequencing was performed using weighted gene co-expression network analysis to reveal the key pathways underlying hypoxia regulation and iron metabolism. The main results are as follows. (1) Except for the hypoxia-inducible factor 1 (HIF-1) content (between the low- and high-iron groups), significant differences were not observed among the serum iron metabolic parameters. The serum HIF-1 content of the low-iron group was significantly higher than that of the high-iron group (*p* < 0.05). (2) Sequencing analysis of the liver transcriptome revealed 155 differentially expressed genes (DEGs) between the low-iron and control groups, 229 DEGs between the high-iron and control groups, and 279 DEGs between the low- and high-iron groups. Bioinformatics analysis showed that the HIF-1 and transforming growth factor-beta (TGF-β) signaling pathways were the key pathways for hypoxia regulation and iron metabolism. Four genes were selected for qPCR validation, and the results were consistent with the transcriptome sequencing data. In summary, the serum iron metabolism parameter results showed that under the influence of low- and high-iron diets, Wujin piglets maintain a steady state of physiological and biochemical indices via complex metabolic regulation of the body, which reflects their stress resistance and adaptability. The transcriptome results revealed the effects of low-iron and high-iron diets on the gene expression level in the liver and showed that the HIF-1 and TGF-β signaling pathways were key for regulating hypoxia adaptability and iron metabolism homeostasis under low-iron and high-iron diets. Moreover, HIF-1α and HEPC were the key genes. The findings provide a theoretical foundation for exploring the regulatory pathways and characteristics of iron metabolism in Wujin pigs.

## 1. Introduction

Iron is an essential trace element for maintaining life and growth, which is widely distributed in the body and participates in several biochemical reactions [1,2]. Iron deficiency or excess affects health, with iron deficiency leading to anemia, fatigue, decreased immunity, and other symptoms, and excessive iron leading to oxidative damage, overactivation of the immune system, organ function damage, and other symptoms. Therefore, the iron content in the body of pigs directly affects their growth, development, and metabolic processes [3,4]. Iron deficiency in piglets reduces immune and antioxidant functions [5], causes iron-deficiency anemia, affects hematopoietic function, and even leads to the death of piglets in severe cases [6]. When supplemented with excessive iron, piglets seem to be unable to effectively metabolize [7] and may experience diarrhea and other symptoms [8]. Supplementing 150 mg Fe/kg to the basic diet of piglets can increase the iron status of tissues and the activity of iron-containing enzymes [9], when the amount of iron added is up to 250 mg/kg, it will increase the incidence rate of diarrhea and the number of intestinal fecal coliforms in piglets [10,11]. Similarly, a high Fe (750 mg/kg) diet resulted in altered iron signaling cascades of ferromodulin and iron transport proteins in calves [12]. Therefore, iron supplementation in weaned piglets has a very important role and significance in pig production. It can promote growth and development, prevent anemia, improve immunity, ensure digestive function, improve pig production efficiency, ensure pork quality, and improve economic benefits.

Serum is the product obtained after removing erythrocytes and platelets from human or animal whole blood after coagulation, and it can reflect the metabolic status of the body to a certain extent. Moreover, it can be used to evaluate the physiological status and diseases of the body. The parameters related to iron metabolism in serum mainly include the serum iron, transferrin (Tf), transferrin receptor (Tfr), total iron binding capacity (TIBC), hemoglobin (Hb), erythropoietin (EPO), ferritin, and hepcidin (HEPC) contents. These indices can comprehensively reflect the body’s iron metabolism and have important clinical value for diagnosing iron metabolism diseases and anemia and evaluating iron nutritional status.

Transcript expression levels are also an important index for judging the metabolic status of the body. The transcriptome refers to the collection of all transcription products in a cell in a broad sense and the collection of all mRNA in a narrow sense [13]. It can reflect the gene expression of an organism in a specific environment and help reveal the transcription regulation mechanisms of organisms in different physiological and pathological states. Thus, this method can provide a deep understanding of the essence of life and the mechanism of disease. For example, Li et al. [14] used dietary interventions in rats with different iron and copper concentrations, and through liver transcriptome analysis, they found that copper supplementation had the potential to alleviate iron-induced hypercholesterolemia. Meena [15] performed a transcriptome analysis of the mechanism of iron deficiency tolerance in hexaploid wheat to select an appropriate genotype to maximize yield. Yang et al. [16] explored the causal relationship between hepatic iron accumulation and lipid metabolism through comprehensive in vitro and in vivo experiments combined with in situ analysis and RNA sequencing to provide new perspectives for understanding the pathogenesis of fatty liver disease and its progression. In recent years, with the development of high-throughput sequencing technology, the use of RNA-Seq, which is the main transcriptome quantitative analysis system, has increased rapidly [17,18]. Many researchers in the biology, biotechnology, and medicine fields have performed transcriptome analyses to study the state, physiology, and activity of cells [9,10]. The sequence of each transcript fragment can be directly determined using RNA-Seq, which is important for the analysis of unknown genes and new transcription isomers [19,20,21,22,23].

Wujin pig is mainly distributed in the Yunnan-Guizhou Plateau at higher altitudes and represents a unique pig breed in the plateau pastoral area of Southwest China [24]. Based on its long-term genetic evolution under a plateau environment, the Wujin pig shows excellent anti-stress ability [25]. Iron regulation is closely related to the development of mechanisms to adapt to hypoxic environments in plateau animals. In our previous research, we used the Yuedawu pigs, which is of the same bloodline as the Wujin pig [26], to conduct nutritional studies, and found that lactoferrin could improve iron nutrition of sows and piglets by increasing the RBC and HGB levels in the blood of sows and piglets, as well as the iron content of sows’ serum and milk [27]. In addition, a study of hypoxia adaptation in Wujin pigs and Yuedawu pigs revealed that the physiological parameters of hypoxia adaptation in Wujin pigs were significantly higher than those in Yuedawu pigs, and the expression of hypoxia adaptation genes in sows was significantly higher than that in boars [24]. However, the intrinsic link between the differences in hypoxia-adapted gene expression and hypoxia-adaptation ability of pigs needs further study, and there are few studies on iron nutrition and its regulatory mechanisms in highland pigs. Therefore, we investigated the mechanism of iron regulation in Wujin pigs by setting different dietary iron levels. Our study provides a theoretical basis for improving nutritional research and resource utilization and exploring the regulatory pathways and characteristics of iron metabolism in Wujin pigs.

## 2. Materials and Methods

### 2.1. Animals and Feeding Conditions

Eighteen healthy 35-day-old Wujin pigs (half male and half female, with similar weights) were randomly divided into three groups: control, low iron, and high iron, six replicates were performed for each group, with one pig in each replicate. The experimental animals were raised at the Yunnan Diandong Hunter Agriculture and Forestry Development Co., Ltd. in Kunming, China, and each piglet was raised separately. The experimental feed was provided twice daily (8:00 a.m. and 5:00 p.m.). Piglets could drink freely, and the farm adopted routine immunization and disinfection procedures. The pre-feeding period was 5 days, and the formal experimental period was 30 days.

### 2.2. Diet Preparation

A basic diet was prepared according to the National Research Council (NRC) Nutrient Requirements of Swine (2012) [28] and the nutritional requirements of pigs (GB/T 39235-2020) [29] (Table 1). The control group received 100 mg/kg iron (provided as iron glycinate) in the basic diet, the low-iron group did not receive additional iron supplementation, and the high-iron group received 200 mg/kg iron (provided as iron glycinate) in the basic diet. Iron glycinate (glycine ≥ 21%, Fe^2+^ ≥ 17%) was purchased from Hangzhou Huineng Animal Medicine Co., Ltd., Hangzhou, China.

### 2.3. Sample Collection

Five-milliliter blood samples (a total of 18) were collected from the anterior vena cava using a blood collection needle (vacuum blood collection tube without anticoagulants). After standing at 20 °C for 1 h, the samples were centrifuged at 3000 rpm for 15 min, and the serum was collected for determining biochemical indexes and stored at −20 °C. Subsequently, 18 piglets were slaughtered and liver samples were collected and put into a freezing tube and then stored at −80 °C. Three RNA samples were randomly selected from each group for subsequent RNA-Seq and qPCR experiments.

### 2.4. Growth Performance

The body weight of each group was recorded on the 0th and 30th day of the test. The daily feed intake and residual feed were also recorded. The formula is as follows:Average daily feed intake (ADFI) = feed consumption in g/day d;
Average daily weight gain (ADG) = (final weight − initial weight) g/day d;
Feed to weight ratio (F/G) = ADFI/ADG;

### 2.5. Analysis of Serum Iron Metabolism Parameters

The total iron binding capacity (TIBC) and iron content were determined using relevant commercial kits (Nanjing Jiancheng Bioengineering Institute, Nanjing, China). Serum levels of hypoxia-inducible factor 1 (HIF-1), HEPC, TfR, HB, and EPO were determined using relevant commercial kits (Shanghai Enzyme-Linked Biotechnology Co., Ltd., Shanghai, China).

### 2.6. RNA Isolation and Sequencing and Sequence Data Processing

#### 2.6.1. RNA Extraction

Three liver samples were randomly selected from each group for RNA extraction using the RNAsimple Total RNA Kit (Tiangen Biotech Co., Ltd., Beijing, China).

#### 2.6.2. Quantitative and Qualitative RNA Analysis

RNA concentration and purity were measured using a NanoDrop 2000 (Thermo Fisher Scientific, Wilmington, DE, USA). RNA integrity was evaluated using an RNA Nano 6000 detection kit for an Agilent Bioanalyzer 2100 System (Agilent Technologies, Santa Clara, CA, USA).

#### 2.6.3. Library Preparation of Transcriptome Sequencing

The total of 1 μg RNA in each sample was used as the input material for RNA sample preparation. According to the manufacturer’s suggestion, a NEBNext Ultra RNA Library Prep Kit (Illumina, Omaha, NE, USA) was used to generate a sequencing library, and the index code was added to determine the sequence of each sample. Briefly, mRNA was purified from the total RNA using magnetic beads attached to poly-T oligonucleotides. Disruption was carried out using divalent cations in the first-chain synthesis reaction buffer (5×) of NEBNext at high temperatures. The first cDNA strand was synthesized using random hexamer primers and M-MuLV reverse transcriptase. Subsequently, DNA polymerase I and RNase H were used to synthesize second-strand cDNA. The remaining overhangs were converted into blunt ends using exonuclease/polymerase. After the 3′ end of the DNA fragment was adenylated, the NEBNext linker with a hairpin loop structure was connected to prepare for hybridization. To select a cDNA fragment with a length of 240 bp, library fragments were purified using the AMPure XP system (Beckman Coulter, Beverly, CA, USA). Prior to PCR, 3 μL of User Enzyme (Neb, Ipswich, MA, USA) was applied at 37 °C for 15 min and then 95 °C for 5 min together with the cDNA connected with the size-selective adaptor. Phusion high-fidelity DNA polymerase, universal PCR primers, and index (X) primers were used for PCR. Finally, the PCR products were purified (AMPure XP system) and the library quality was evaluated using the Agilent Bioanalyzer 2100 system(Agilent Technologies, Santa Clara, CA, USA).

#### 2.6.4. Clustering and Sorting

The TruSeq PE Cluster Kit v4-cBot-HS (Illumina, Omaha, NE, USA) was used to cluster index-coded samples on the cBoT cluster generation system according to the manufacturer’s instructions. After cluster generation, the library was sequenced on an Illumina platform, and paired terminal readings were generated.

#### 2.6.5. Quality Control

Raw data in FASTQ format (raw reading) were first processed using a Perl script. Clean data were obtained by removing readings containing adapters, poly-N, and low-quality readings from the original data. Simultaneously, the Q20, Q30, and GC content and sequence repetition levels of the clean data were calculated. All downstream analyses were based on clean high-quality data.

#### 2.6.6. Contrast Ratio Analysis

Connectors and low-quality sequence reads were removed from the dataset. After data processing, the original sequences were converted to a clean read. These reads were plotted against the reference genome sequence (Sscrofa11.1_ncbi). Hisat2 2.0.1 software was used to compare the reference genomes [30] and StringTie comparisons were performed to assemble the reads [31].

#### 2.6.7. Gene Function Annotation

Gene functions were annotated based on the following databases: Nr (NCBI non-redundant protein sequences), Nt (NCBI non-redundant nucleotide sequences), Pfam (protein family), KOG/COG (Clusters of Orthologous Groups of proteins), Swiss-Prot (a manually annotated and reviewed protein sequence database), KO (KEGG Ortholog database), and GO (Gene Ontology).

### 2.7. Bioinformatics Analysis

#### 2.7.1. Gene Expression Level Quantification, Differential Gene Screening, and Functional Enrichment and Cluster Analyses

Fragments per kilobase of transcript per million fragments mapped (FPKM) were used as an index to measure the expression levels of transcripts or genes [32]. Based on the number of genes in each sample, differentially expressed genes (DEGs) were screened using DESeq2 1.44.0 software [33]. Fold change ≥ 2 and *p* < 0.05 values were used as the screening criteria for DEGs. Kyoto Encyclopedia of Genes and Genomes (KEGG) and GO function enrichment analyses of the screened differentially expressed genes were carried out to determine the biological function or metabolic pathways mainly affected by the DEGs. In this study, three comparison groups were set up: A (low-iron group vs. control group), B (high-iron group vs. control group), and C (high-iron group vs. low-iron group).

#### 2.7.2. Hypoxia Regulation and Screening of Key Signal Pathways of Iron Metabolism

Weighted gene co-expression network analysis (WGCNA) indicated that the gene network obeys a scale-free distribution. In addition, the correlation matrix of gene co-expression and adjacency function formed by the gene network were defined and the correlation coefficients of different nodes were calculated. Based on these results, a hierarchical clustering tree was constructed. Different branches of the clustering tree represent different gene modules. The degree of co-expression of genes in a module is high, whereas the degree of co-expression of genes belonging to different modules is low. Subsequently, the relationship between modular genes and apparent serum traits was explored, and genes related to hypoxia regulation and iron metabolism were screened. GO and KEGG enrichment analyses were carried out on the screened genes, and the obtained pathways were compared with the pathways enriched by differential genes. The main signal pathways related to hypoxia regulation and iron metabolism affected by dietary iron level in this study were screened out.

### 2.8. Gene Expression

Total RNA (18 samples) was extracted using the RNAsimple Total RNA Kit (Tiangen Biotech Co., Ltd., Beijing, China) and the first strand cDNA was synthesized using the FastKing cDNA synthesis kit (Tiangen Biotech Co., Ltd., Beijing, China). Fluorescent quantitative PCR was performed using a SuperReal fluorescent quantitative premix kit (SYBR Green, Tiangen Biotech Co., Ltd., Beijing, China). The primers (hepcidin (*HEPC*), erythropoietin (*EPO*), hypoxia-inducible factor 1 subunit alpha (*HIF-1α*), transferrin receptor (*TFRC*)) are shown in Table 2. β-actin (*ACTB*) was selected as the housekeeping gene. According to the expression of the β-actin gene, the expression level of mRNA was standardized, and the relative expression levels were calculated by using the 2^−∆∆Ct^ method [34].

### 2.9. Statistical Analysis

All data are presented as the mean ± standard deviation (SD). The results were considered statistically significant at *p* < 0.05. All statistical analyses were performed using SPSS v21.0. Data were analyzed using one-way ANOVA, and multiple comparisons were performed using Duncan’s test.

## 3. Results

### 3.1. Effect of Different Dietary Iron Contents on the Growth Performance of Piglets

The results shown in Table 3 indicate that as the dietary iron content increases, the feed-to-weight ratio gradually decreases.

### 3.2. Effects of Different Dietary Iron Contents on Serum Iron Metabolism Parameters of Piglets

According to Table 4, the serum HIF-1 content in the low-iron group was significantly higher than that in the high-iron group (*p* < 0.05). There were no significant differences in the total iron-binding capacity or the iron, HEPC, TfR, HB, and EPO contents among the three groups (*p* > 0.05).

### 3.3. Sequencing Results and Quality Control

Three liver tissue samples from each group were sent to Beijing Baimaike Biotechnology Co. Ltd. for transcriptome sequencing. Table 5 shows that a total of 57.35 Gb clean data were obtained in this sequencing. The clean data of each sample reached 5.71 Gb, and the base percentage of Q30 was 93.17% or higher. The clean reads of each sample were compared with the designated reference genome (Sscrofa11.1), and the mapping efficiency ranged from 96.22% to 97.03%. The proportion of clean reads with genes located in each sample was >93%. The proportion of clean reads relative to the intergenic region was less than 7%, and the part of the sequence that was not filtered after strict quality control may be a new non-coding RNA or a transcription self-correcting gene in Wujin pig. These results demonstrate that the transcriptome sequencing data are reliable and suitable for further analysis.

### 3.4. Screening of Differentially Expressed Genes

In this study, 25,918 genes were obtained by comparison with the database and annotation, and 5258 new genes were identified, of which 984 were annotated. Meanwhile, three control DEGs were screened based on the sequencing results. The numbers of differential genes obtained from the comparisons are shown in Table 6 and Figure 1. A total of 577 DEGs were detected among the three groups (|fold change| ≥ 2, *p* < 0.05). A total of 155 differentially expressed genes were identified between the low-iron and control groups, including 78 upregulated and 77 downregulated genes; 229 differentially expressed genes were identified between the high-iron group and control group, including 164 upregulated and 65 downregulated genes; and 279 differentially expressed genes were identified between the high- and low-iron groups, including 165 upregulated and 114 downregulated genes.

### 3.5. Functional Enrichment of DEGs

GO function annotation and KEGG pathway enrichment analyses were performed on the three groups of DEGs. The results of the GO classification map of the low-iron and control groups are shown in Figure 2. A total of 155 DEGs were classified into 42 categories, including 19 “biological processes”, 14 “cellular components” and 9 “molecular functions”. As shown in Figure 3, 155 DEGs were significantly enriched in 18 KEGG pathways, and the differential genes were mainly enriched in pyrimidine metabolism, peroxisome proliferator-activated receptor (PPAR) signaling pathway, fructose and mannose metabolism, biosynthesis of unsaturated fatty acids, and the HIF-1 signaling pathway related to iron metabolism, which was upregulated.

The results of the GO classification map for the high-iron group and control groups are shown in Figure 4. A total of 229 DEGs were classified into 45 categories, including 19 “biological processes”, 14 “cellular components” and 12 “molecular functions”. Among them, the antioxidant activity related to iron nutrition was enriched in molecular functions. In total, 229 DEGs were significantly enriched in 53 KEGG pathways. Figure 5 shows the top 20 upregulated pathways and 15 downregulated pathways. Differential genes were mainly enriched in the NF-kappa B signaling pathway, B cell receptor signaling pathway, protein digestion and absorption, and iron metabolism. Metabolism was related to hematopoietic cell lineage, HIF-1 signaling, PI3K-Akt signaling, and NF-κB signaling pathway.

The results of the GO classification map for the high- and low-iron groups are shown in Figure 6. A total of 279 DEGs were classified into 47 categories, including 20 “biological processes”, 17 “cellular components” and 10 “molecular functions”. In total, 279 DEGs were significantly enriched in 39 KEGG pathways. Figure 7 shows the top 20 upregulated and 7 downregulated channels. Differential genes were mainly enriched in the NF-kappa B signaling pathway, B cell receptor signaling pathway, protein digestion and absorption, hematopoietic cell lineage, PI3K-Akt signaling pathway, ovarian steroidogenesis, PI3K-Akt signaling pathway, and the HIF-1 signaling pathway.

### 3.6. Screening of Hypoxic Regulation- and Iron Metabolism-Related Pathways Based on WGCNA and Pathway Enrichment Analyses

Sample cluster analysis was performed according to the expression patterns of all genes, and the results are shown in Figure 8a. The visualization results of the co-expression network are shown in Figure 8b. The genes corresponding to the same color on the cluster belong to the same module.

The correlation thermogram between the gene module and serum biochemical indexes (Figure 9) shows that there is a high correlation between the genes in the MEgreen and the content of TfR (*p* = 0.06, |R| = 0.65). Genes in the MEred were highly correlated with the EPO content (*p* = 0.09, |R| = 0.6) and significantly correlated with HIF-1 content (*p* = 0.001, |R| = 0.89).

KEGG enrichment analysis was carried out on the genes in the MEgreen and MEred (Figure 10). The green module showed enriched signal pathways related to hypoxia and iron regulation, including the TGF-beta signaling, hematopoietic cell lineage, NF-kappa B signaling, and PI3K-Akt signaling pathways. The signaling pathways related to hypoxia and iron regulation enriched by MEred included the HIF-1 signaling, hematopoietic cell lineage, and PI3K-Akt signaling pathways.

### 3.7. Relative Expression Analysis of Hypoxia Regulation and Iron Metabolism Gene mRNA in Piglets

In this study, four genes (*HEPC*, *HIF*-*1α*, *EPO*, and *TFRC*) involved in the HIF-1 and TGF-beta signaling pathways were further screened via real-time fluorescence quantitative PCR, and the results are shown in Figure 11. Among the above genes, in the low-iron group, the expression of *HEPC* and *HIF*-*1α* was significantly lower while *TFRC* and *EPO* expression was significantly higher than that of the control group, which was consistent with the fluorescence quantitative PCR results. In the high-iron group, significant differences were not observed in *HEPC* and *EPO* expression, and significant decreases were observed in *HIF*-*1α*, while significant increases were observed in *TFRC* expression compared with the control group, which were consistent with the results of fluorescence quantitative PCR. After testing, the differential expression of these genes was consistent with the results of transcriptome sequencing, thus indicating the accuracy of the sequencing data and reliability of the channels and genes screened in this study.

## 4. Discussion

### 4.1. Growth Performance

Newborn piglets are mainly dependent on breast milk for energy and nutrition, and in this context, sow’s milk is an important source of micronutrients for piglets. In general, newborn piglets may experience iron deficiency, which may lead to iron deficiency anemia. Compared with healthy piglets, iron-deficient piglets have a slower growth rate and require timely supplementation of iron [35]. As the piglets grow, the iron in the sow’s milk is not sufficient to meet their growth demands; consequently, exogenous sources of iron are needed after weaning. In pig farming, it is routine and mandatory to supplement piglets with different forms of iron at different doses and times of the year [36,37]. However, iron supplementation in weaned piglets is not fully effective for growth performance [38], consistent with our iron supplementation in Wujin piglets.

### 4.2. Changes in Serum Iron Metabolism Parameters

The serum contains various physiologically active substances, such as immunoglobulins, and serum parameters reflect the metabolic status of the body and can be used to diagnose diseases, monitor therapeutic effects, and evaluate health status. In this study, the serum iron, TIBC, TfR, HB, HEPC, HIF-1, and EPO contents of Wujin pigs were determined. HIF-1 is a key factor in hypoxia regulation, and its secretion is regulated by oxygen partial pressure and downstream effects regulate the expression of EPO and TFRC. The contents of HIF-1 and EPO in serum can reflect the degree of hypoxia stress [39]. Under hypoxic conditions, the secretion of HIF-1 and EPO increases [24,40]; an acute hypoxia treatment was first applied to rats, which found that the contents of HIF-1 and EPO in the serum increased significantly. Subsequently, they determine the level of hypoxia adaptability through intermittent hypoxia treatments. After applying the same hypoxia treatment, the serum HIF-1 and EPO levels in rats still increased but were significantly lower than those in the acute hypoxia group, which demonstrated that with the enhancement of hypoxia adaptability, the changes in metabolic level can also be alleviated. In this study, the serum HIF-1 content in the control and high-iron groups showed a downward trend compared with that in the low-iron group. The decline in the high-iron group was more pronounced, indicating that iron in the diet alleviated the regulatory mechanisms stimulated by hypoxia in Wujin pigs.

The above results show that the dietary iron level can affect the physiological state of iron metabolism in Wujin pigs; however, it was still within the adaptive range of Wujin pigs.

### 4.3. Transcriptional Analysis

A transcriptome is a complete set of transcripts in a cell or tissue at a specific developmental stage or under physiological conditions. Understanding the transcriptome is important for explaining the functional elements of the genome, revealing the molecular components of cells and tissues, and understanding development and diseases [41]. With the development of high-throughput sequencing technology, RNA-Seq has become the main method for transcriptome sequencing because of its low cost, high sensitivity, and large dynamic detection range [42]. In the present study, the sequencing quality of the liver transcriptome was good and can be further analyzed. The DEGs in the three control groups were subjected to GO function and KEGG pathway enrichment analyses. In the GO classification and enrichment analyses, antioxidant activity related to iron nutrition was enriched in molecular functions in control group B, indicating that high levels of iron in the feed would affect the antioxidant capacity of Wujin pigs, which may be due to the unique physical and chemical properties of iron [43].

In the KEGG pathway enrichment analysis, most of the enriched pathways were involved in essential life activities and growth and development functions of piglets. For example, the upregulated DEGs in control group A were also enriched in PPAR signaling, biosynthesis of unsaturated fatty acids, and fatty acid metabolism pathways, among which PPAR is a nuclear hormone receptor activated by fatty acids and their derivatives that can regulate lipid metabolism and fat formation and maintain the expression of genes involved in metabolic homeostasis and inflammation. Downregulated DEGs were enriched in the AGE-RAGE, thyroid hormone, and relaxin signaling pathways, which are responsible for regulating glucose metabolism and coping with oxidative stress-induced damage [44,45]. This shows that a low-iron diet affects lipid and glucose metabolism in Wujin pigs and affects the body’s resistance to oxidative stress. In contrast group B, the upregulated differential genes are enriched in hematopoietic cell lineage, NF-κB signaling pathway, and HIF-1 signaling pathway related to hematopoietic and hypoxia regulation, which are mainly responsible for regulating oxygen transport, angiogenesis, and inflammatory response. Downregulated genes were enriched in regulatory pathways, such as chemical carcinogenesis and cancer pathways. This indicates that a high-iron diet affects the transport and utilization of oxygen in the body and regulates diseases. Simultaneously, enriched signaling pathways were screened according to their functions. The signaling pathway related to hypoxia regulation and iron metabolism in control group A was HIF-1, while those related to hypoxia regulation and iron metabolism in control group B were NF-κB, hematopoietic cell lineage, PI3K-Akt, and HIF-1. The functions of these pathways and the enrichment sites of different genes were further analyzed in a follow-up study.

### 4.4. Screening Key Signaling Pathways for Hypoxia Regulation and Iron Metabolism Based on WGCNA and Transcriptome Analysis

WGCNA is the primary method used to construct gene coexpression networks. In recent years, many studies have explored the relationship between specific physiological conditions, diseases, cancers, and gene expression through WGCNA and evaluated the synergistic effect and regulatory mechanism between genes [46,47,48,49,50,51]. The results obtained using WGCNA were consistent with the DEG enrichment and screening results, indicating that the steady state of apparent serum parameters in Wujin pigs is closely related to the regulation of the iron metabolism signaling pathway under the influence of different dietary iron levels.

The function of different signaling pathways and the specific sites of gene enrichment were then analyzed. The TGF-β signaling pathway regulates many biological functions. *HEPC*, a DEG in this study, is enriched in the BMP/SMAD/HEPC regulatory pathway, which is a key factor for maintaining iron homeostasis in the body. Hematopoietic cell lineage is a signaling pathway that regulates the differentiation of hematopoietic stem cells into different mature cell lines. In this study, the DEGs *IgM* and *IgD* were enriched in the immunoglobulin regulation pathway. The NF-κB signaling pathway is a typical pro-inflammatory signaling pathway, and NF-κB is an upstream factor of *HIF*-*1α*, which can inhibit its transcription activity by binding to the N-terminal of *HIF*-*1α*. *Bcl*-*2* (DEG) was found to be enriched in the regulatory survival pathway. The PI3K-Akt pathway is a multifunctional signaling pathway that regulates cell growth and metabolism. At the same time, PI3K and Akt are upstream factors of *HIF-1α* and thus can regulate the transcription level of *HIF*-*1α*. *BCR* was enriched in the activation pathway of PI3K, whereas *Bcl-2* and *NUR77* were enriched in the regulation pathway of survival. The HIF-1 signaling pathway is the key pathway of hypoxia regulation. In the present study, *EPO* was found to be enriched during erythropoiesis. *TFRC* was enriched in iron metabolism. *PAI*-*1*, *ANGPT*, and *TIMP*-*1* were enriched during angiogenesis. *PFK2* and *LDHA* were enriched in promoting anaerobic metabolism. *GF* and *HIF*-*1α* were enriched in the *HIF-1α* regulatory pathway. It can be seen that the iron content in diet has a significant impact on the metabolic regulation of the body, and thus understanding the genetic and metabolic responses of dietary changes is crucial [52,53]. According to the results of WGCNA and the enrichment of differentially expressed genes, TGF-β signaling pathways and HIF-1 signaling pathways were selected as the key pathways of hypoxia regulation and iron metabolism, and these two pathways were further analyzed and verified in this study.

### 4.5. Effects of Dietary Iron Level on HIF-1 Signaling Pathway and TGF-β Signaling Pathway in Wujin Piglets

The HIF-1 signaling pathway is the key pathway of hypoxia regulation, and the expression of *HIF*-*1α* is regulated by *NF*-*kβ*, *PI3K*, *AKT*, and *mTOR* in the upstream. In addition, the expression level of *HIF*-*1α* will affect the downstream expression of *EPO*, *TF*, and *TFRC*. In this study, the expression levels of *HIF*-*1α*, *EPO*, and *TFRC* in the HIF-1 signaling pathway were verified by qPCR, and the findings were consistent with the results of transcriptome sequencing.

*HIF*-*1α* encodes the HIF-1α protein, which is a transcription factor activated in hypoxic environments. The content and activity of the HIF-1α protein are affected by oxygen levels. When cells are in a hypoxia environment, the content and activity of the HIF-1α protein will increase, which will activate the transcription and translation of downstream genes regulated by HIF-1α, thus promoting the adaptation and regulation of the body to hypoxia. The mutation and abnormal expression of *HIF*-*1α* are related to the occurrence and development of many diseases, such as tumors, cardiovascular disease, and diabetes [54]. In this study, the expression level of *HIF*-*1α* in the low-iron group and the high-iron group was significantly lower than that in the control group, and that in the low 0 iron group was significantly lower than that in the high-iron group. Thus, the expression level of *HIF*-*1α* was inhibited at low and high iron levels. However, the source of this inhibition needs to be further determined.

The expression level of *EPO* has important reference significance for hypoxia adaptability, anemia treatment, and muscle growth [55]. For example, the expression level of *EPO* in pigs usually increases significantly in hypoxic environments, which promotes the production and release of red blood cells and increases the adaptability of the body to hypoxia [56]. Studies have found that *EPO* expression in Tibetan pigs at high altitudes remains high for a long time and is significantly increased in Yorkshire pigs under hypoxia stress. However, the high expression under this stress state is reduced after the body adapts to a period, indicating that *EPO* expression regulation is a physiological adjustment process for the body to adapt to hypoxic environments [57]. In this study, *EPO* expression in the low-iron group was significantly higher than that in the control group but did not significantly differ between the high-iron group and control group, which means that the low-iron group had more obvious reactions to hypoxia and iron deficiency than the control and high-iron groups.

The expression level of *TFRC* is one of the indices used to evaluate iron metabolism in the body [58]. The expression level of *TFRC* is regulated by intracellular iron content. When the intracellular iron content decreases, the transcription and expression of *TFRC* increase, thereby increasing the absorption and utilization of iron by cells. In contrast, when the iron content in cells is too high, the expression level of *TFRC* decreases, thereby reducing the absorption and utilization of iron by cells and maintaining iron homeostasis [59]. For example, in patients with iron-deficiency anemia, the expression level of *TFRC* is usually high, whereas in patients with iron overload, the expression is low. In this study, *TFRC* expression in the low- and high-iron groups was significantly higher than that in the control group, indicating that liver cells in the low- and high-iron groups required more iron.

The TGF-β signaling pathway is an important signaling pathway that regulates cell proliferation, differentiation, apoptosis, embryonic development, immune regulation, and tissue repair. The BMP-SMAD-HEPC regulatory pathway is closely related to iron metabolism. HEPC is a key factor in iron absorption and metabolic regulation, and it is mainly expressed in the liver but also expressed in small amounts in the heart and spinal cord [60]. *HEPC* controls systemic iron homeostasis by negatively regulating FPN1 [45,46]. HEPC expression increases when the body is overloaded with iron, and iron release into circulation can be inhibited by regulating the homologous receptor FPN1 after translation [61,62,63]. At the molecular level, HEPC regulates cellular iron output by binding to FPN1, thus leading to the ubiquitination and internalization of receptors and subsequent lysosomal degradation in vitro [64] HEPC is secreted quickly and has a half-life of only a few minutes. Therefore, changing the production rate of HEPC can quickly change its circulating concentration in the body, thus changing the speed at which iron enters the blood circulation [65]. Current research shows that the content of HEPC is mainly regulated at the transcription level by factors that include plasma iron concentrations, liver iron reserves, inflammatory reactions, and erythropoiesis [66]. In this study, the expression level of *HEPC* in the low-iron group was significantly lower than that in the high-iron and control groups. The low expression of *HEPC* in the low-iron group indicated that the regulation of the body promoted the release of iron from the intestinal tract and other tissues and cells into the circulatory system and slightly increased the iron content in the serum to maintain iron homeostasis in the body. There was no significant difference in the expression level of *HEPC* between the high-iron and control groups, which may have been caused by multiple regulatory pathways. Although the animals in the high-iron group were in a state of relative iron deficiency, it was necessary to reduce the expression of *HEPC* to promote the release of iron. However, too high an intestinal iron content would stimulate the body to upregulate the expression of HEPC, thus inhibiting the process of iron transport from intestinal epithelial cells to the blood, and the expression level of *HEPC* would be relatively stable under the control of both. In contrast, the slightly lower serum iron content of piglets in the high-iron group may have been caused by lower iron absorption in the intestine, although this requires further verification.

In summary, the three groups of Wujin pigs maintained iron homeostasis and adaptability under hypoxic conditions. The transcriptomics analyses revealed that this is because the body maintains its iron homeostasis and hypoxia adaptability through a series of complex regulatory pathways, among which the HIF-1 signaling and TGF-β signaling pathways are effective regulatory pathways.

## 5. Conclusions

A high-iron diet can relieve hypoxic stress in Wujin piglets. In addition, transcriptomic analyses revealed the main iron homeostasis pathway (TGF-β signaling pathway) and hypoxia regulation pathway (HIF-1 signaling pathway) in Wujin piglets. HEPC was a key factor in iron homeostasis, and HIF-1 was the key factor in hypoxia regulation. In addition, transcription levels were affected by the dietary iron content, and the effect of low dietary iron was more obvious than that of high dietary iron.

## Figures and Tables

**Figure 1 animals-14-02399-f001:**
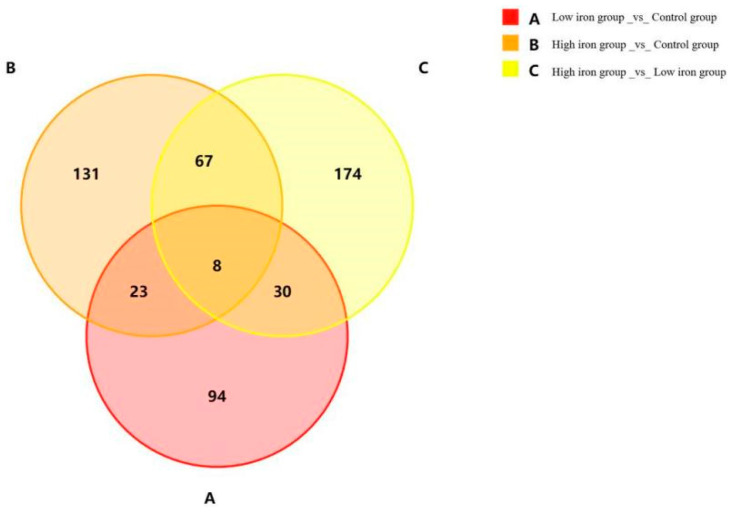
Venn diagram of DEGs in three comparison groups.

**Figure 2 animals-14-02399-f002:**
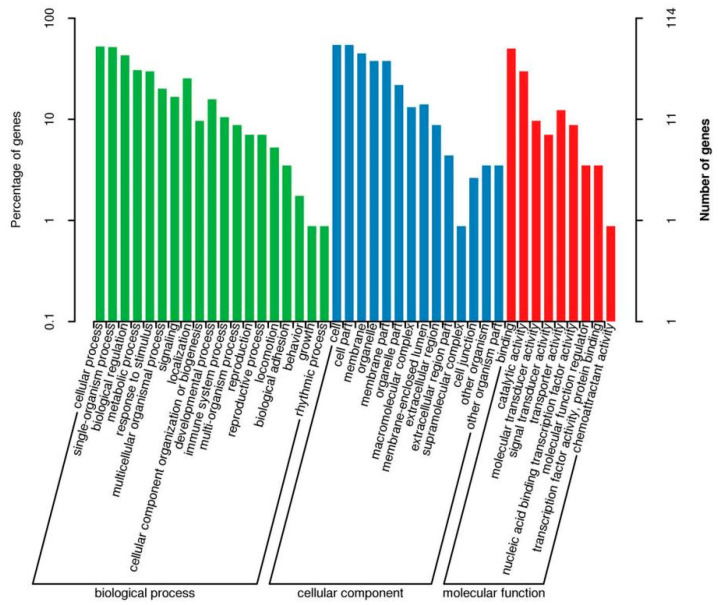
GO enrichment map of differential genes between the low-iron group and control group.

**Figure 3 animals-14-02399-f003:**
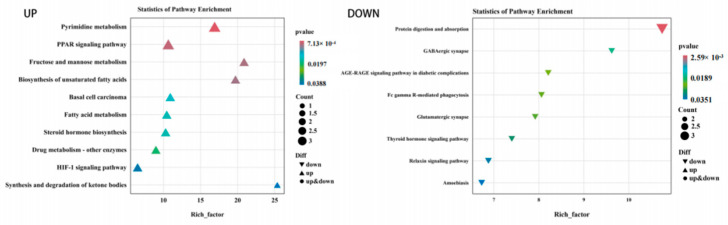
KEGG enrichment map of differential genes between the low-iron group and control group.

**Figure 4 animals-14-02399-f004:**
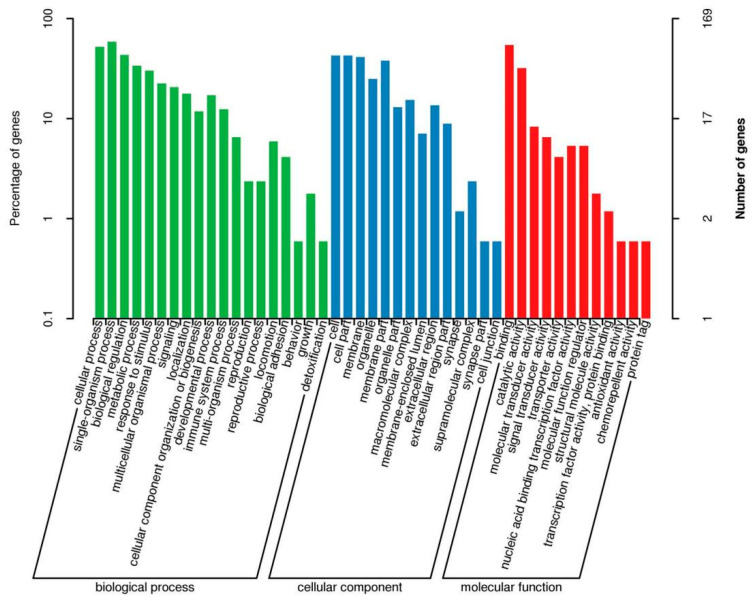
GO enrichment map of differential genes between the high-iron group and control group.

**Figure 5 animals-14-02399-f005:**
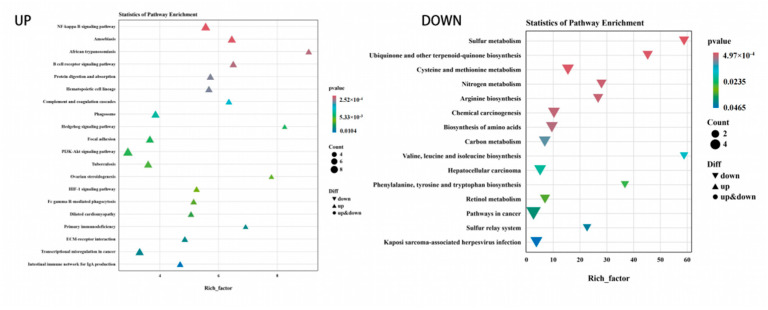
KEGG enrichment analysis diagram of differential genes between the high-iron group and control group.

**Figure 6 animals-14-02399-f006:**
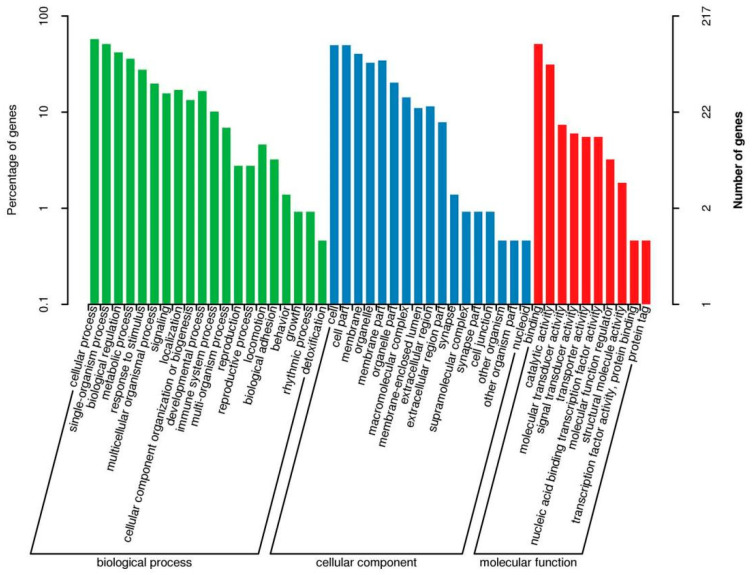
GO enrichment map of differential genes between the low- and high-iron groups.

**Figure 7 animals-14-02399-f007:**
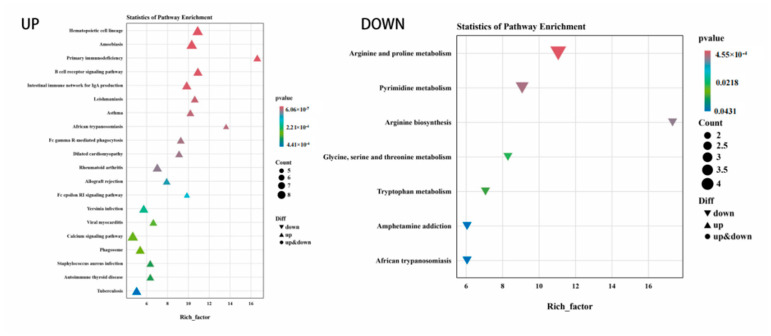
KEGG enrichment analysis diagram of differential genes between the low- and high-iron groups.

**Figure 8 animals-14-02399-f008:**
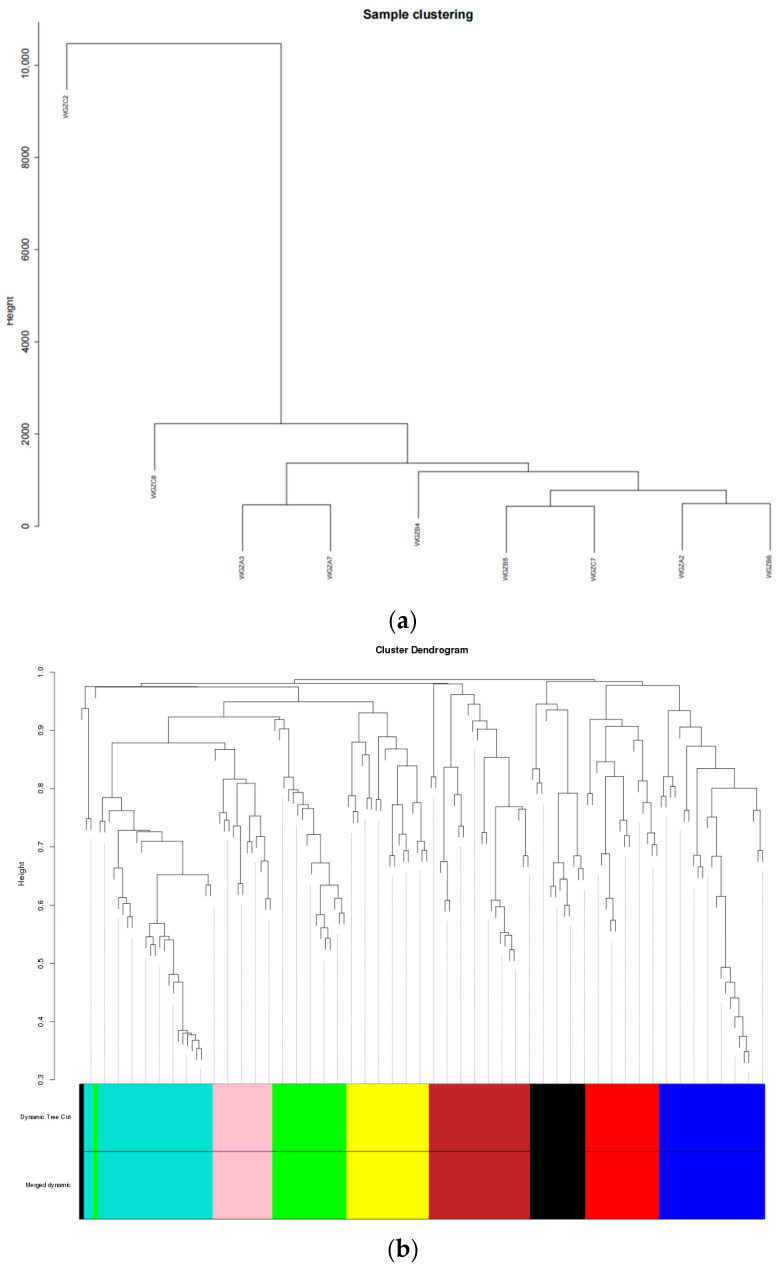
Schematic diagram of sample clustering and co-expression network. (**a**) The results of sample clustering based on Euclidean distance; (**b**) Collaborative expression network visualization.

**Figure 9 animals-14-02399-f009:**
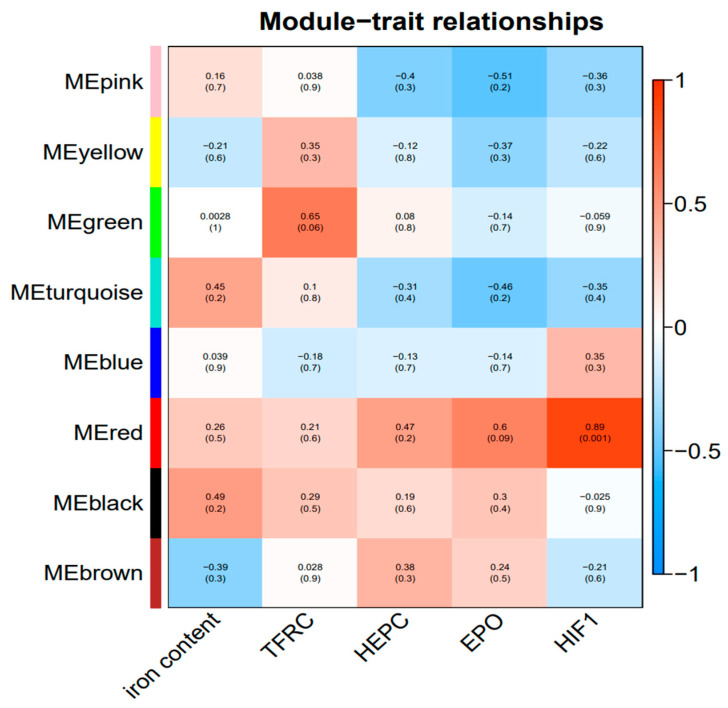
Correlation thermogram between gene modules and traits.

**Figure 10 animals-14-02399-f010:**
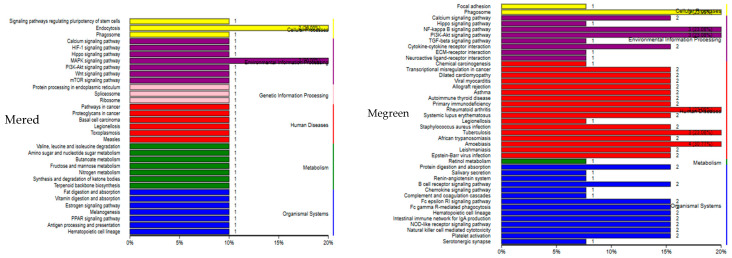
KEGG enrichment results of Red and Green module genes.

**Figure 11 animals-14-02399-f011:**
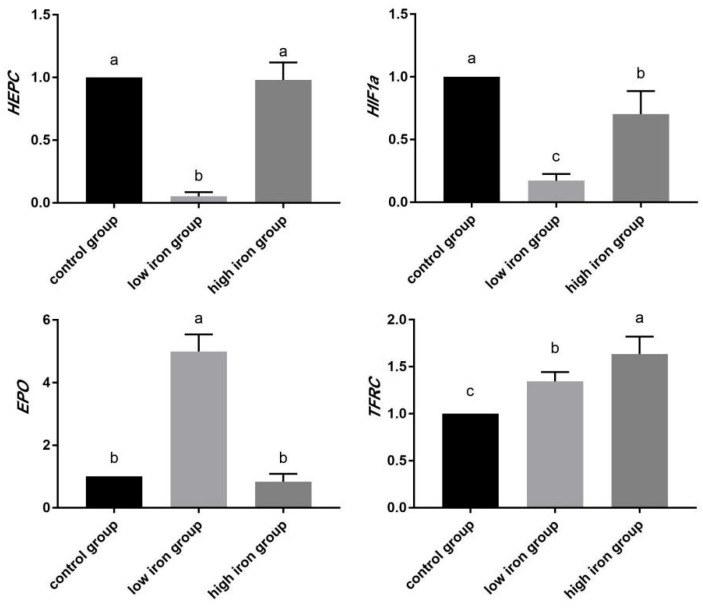
The results of qPCR, a, b, c (*p* < 0.05).

**Table 1 animals-14-02399-t001:** Diet formula and nutrition level.

Ingredient	Content (%)	Nutrient Level ^2^	
Corn	30.0	CP (%)	19.0
Soybean meal	15.0	Ca (%)	0.80
Expanded soybean	15.0	Fe (mg/kg)	76.8
Whey powder	12.0	AP (%)	0.42
Rice	10.0	Lys (%)	1.22
Flour	6.0	Met (%)	0.39
Glucose	4.0	Methionine + cystine (%)	0.67
Peruvian fish meal	3.0	Thr (%)	0.77
Citric acid	1.50	Trp (%)	0.23
Vegetable oil	1.00	DE (MJ/kg)	13.38
Fine stone powder	0.84		
Calcium hydrogen Phosphate	0.81		
Lysine	0.18		
Fungicide	0.10		
DL-methionine	0.08		
Premix ^1^	0.49		

^1^ Premix provides the following nutrients (per kg): VA, 100 mg; VD3, 100 mg; VE, 4400 mg; VK, 100 mg; VB1, 100 mg; VB2, 300 mg; Cu, 10 mg; Zn, 110 mg; Mn, 30 mg; I, 0.2 mg; Se, 0.3 mg; VB3, 1000 mg; VB5, 800 mg; VB12, 500 mg; biotin, 1500 mg; folic acid, 100 mg; and zeolite powder, 298.73 g. ^2^ Fe was calculated, other items were measured.

**Table 2 animals-14-02399-t002:** PCR primer sequences.

Genes	Accession	Primer Sequence (5′-3′)	Amplification Length/bp
*HEPC*	NM_214117	F: AGAGGCTAAGAAGAGACA	76
		R: ATCCCACAGATTGCTTTG	
*EPO*	NM_214134	F: TGCAGCTTCAGTGAGAATATC	75
		R: CTGGACCTCCATCCTCTT	
*HIF-1α*	NM_001123124	F: TACAACATCACCATACAG	118
		R: GACAGATAACACATTAGGA	
*TFRC*	NM_214001	F: CACAAGGAGTAACCACAT	117
		R: GCCAAGTAGCCAATCATA	
*ACTB*	XM_003124280.5	F: GCGGCATCCACGAAACTA	75
		R: TGTTGGCGTAGAGGTCCTT	

**Table 3 animals-14-02399-t003:** Effects of different dietary iron contents on the growth performance of Wujin piglets.

Item	Low-Iron Group	Control Group	High-Iron Group
IBW (kg)	15.21 ± 1.81	15.1 ± 1.93	15.56 ± 1.70
FBW (kg)	27.43 ± 5.33	28.79 ± 4.57	28.56 ± 3.06
ADG (kg/d)	0.38 ± 0.11	0.43 ± 0.10	0.41 ± 0.08
ADFI (kg/d)	1.04 ± 0.24	1.04 ± 0.20	1.00 ± 0.17
F/D	2.81 ± 0.45	2.48 ± 0.36	2.50 ± 0.25

**Table 4 animals-14-02399-t004:** Effects of different dietary iron contents on the serum iron metabolism parameters of Wujin piglets.

Item	Low-Iron Group	Control Group	High-Iron Group
Serum iron content (mg/L)	19.79 ± 2.35	18.51 ± 1.33	16.17 ± 1.88
TIBC (µmol/L)	135.96 ± 6.62	143.63 ± 6.80	142.19 ± 9.46
TfR (nmol/L)	16.01 ± 0.68	16.65 ± 0.27	17.15 ± 0.69
HB (µg/mL)	63.42 ± 6.71	61.41 ± 1.79	58.33 ± 4.44
HEPC (ng/mL)	1135.86 ± 54.75	1179.33 ± 22.76	1108.57 ± 30.38
HIF-1 (pg/mL)	447.67 ± 37.95 ^a^	394.5 ± 13.87 ^ab^	370.52 ± 16.31 ^b^
EPO (mIU/mL)	4.98 ± 0.29	4.89 ± 0.10	4.51 ± 0.19

^a,b^ The same lowercase letters represent no significant differences, whereas different lowercase letters represent significant differences (*p* < 0.05).

**Table 5 animals-14-02399-t005:** Sequencing data statistics and comparison results.

Samples	Clean Reads	Clean Bases	GC Content (%)	% ≥ Q30 (%)	Total Reads	Mapped Reads	Mapping (%)
WGZA2	19,088,882	5,715,504,106	49.33%	94.05%	38,177,764	36,841,761	96.50%
WGZA3	22,129,212	6,623,511,384	49.78%	93.17%	44,258,424	42,654,731	96.38%
WGZA7	19,728,590	5,898,720,736	49.43%	94.35%	39,457,180	38,023,783	96.37%
WGZB4	19,980,454	5,981,164,218	49.48%	94.53%	39,960,908	38,591,996	96.57%
WGZB5	20,097,375	6,016,690,334	48.62%	93.81%	40,194,750	38,823,158	96.59%
WGZB6	19,062,074	5,707,294,314	49.53%	93.77%	38,124,148	36,698,850	96.26%
WGZC2	26,959,352	8,070,051,376	49.73%	93.78%	53,918,704	52,099,313	96.63%
WGZC7	20,765,796	6,210,772,130	49.66%	94.17%	41,531,592	39,963,275	96.22%
WGZC8	23,796,379	7,123,213,814	49.34%	95.17%	47,592,758	46,181,559	97.03%

**Table 6 animals-14-02399-t006:** Statistics on the number of differentially expressed genes.

DEG Set	DEG Number	Upregulated	Downregulated
Low iron group _vs._ Control group	155	78	77
High iron group _vs._ Control group	229	164	65
High iron group _vs._ Low iron group	279	165	114

## Data Availability

Raw reads of transcriptome sequencing of the liver are available at GEO. To review GEO accession GSE247384: Go to https://www.ncbi.nlm.nih.gov/geo/query/acc.cgi?acc=GSE247384 (accessed on 17 April 2024) and the token is udwjewcahbgjbsf.

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
