# Peer review of "Effect of Different Dietary Iron Contents on Liver Transcriptome Characteristics in Wujin Pigs"

_animals, 2024, doi:10.3390/ani14162399_

Round 1

Reviewer 1 Report

Comments and Suggestions for Authors

Iron deficiency in pig nutrition has been a relevant issue for many years. Therefore, the presented study on identifying genes that regulate physiological processes related to iron metabolism in pigs is extremely important. The presented study undoubtedly contains new knowledge in this field and holds great promise for application in the development of pig feeding programs.

The study deserves high praise, with a few minor suggestions:

1. In line 92, if there is data (from other scientific sources) on the iron content levels in pigs of other breeds raised in China, in order to emphasize the physiological characteristics of Wujin pigs.

2. In the caption, provide the results of transcriptome characteristics research at different levels of iron-containing diets in pigs or other animal species.

3. In Table 6, replace "Upregulated" with "Downregulated" in the second case.

4. In the discussion, slightly shorten the repetition of research results and consider comparing them with the results of other scientific works in the field of iron nutrition or other nutrients, highlighting similarities or differences that can help better understand the regulation patterns of various processes through transcriptional analysis.

Author Response

Comment 1: [In line 92, if there is data (from other scientific sources) on the iron content levels in pigs of other breeds raised in China, in order to emphasize the physiological characteristics of Wujin pigs.

Response 1: [Thank you for your question. We think this suggestion will be very helpful to improve this paper. In the article "Study on the differences in hypoxia adaptation between Yunnan Wujin and Yue Crow Wu pigs", the differences in iron and iron metabolism between Wujin pigs and other pig breeds were compared, and the relevant content was added to the introduction.

Comment 2: [ In the caption, provide the results of transcriptome characteristics research at different levels of iron-containing diets in pigs or other animal species.

Response 2: [There are fewer studies on the transcriptome of pigs receiving iron-containing diets, but we have provided some iron-related transcriptome data in the introduction for reference.

Comment 3: [ In Table 6, replace "Upregulated" with "Downregulated" in the second case.

Response 3: [We apologize for the oversight. We have corrected the issue and highlighted it in blue.

Comment 4: [In the discussion, slightly shorten the repetition of research results and consider comparing them with the results of other scientific works in the field of iron nutrition or other nutrients, highlighting similarities or differences that can help better understand the regulation patterns of various processes through transcriptional analysis.

Response 4: [Thank you for your suggestion. We have made changes to the text and marked them in blue.

Reviewer 2 Report

Comments and Suggestions for Authors

This study assessed the effects of different dietary iron levels on serum iron metabolism parameters of Wujin pigs. The experimental design was as follows: 3 groups of pigs were formed, which had different diets (low iron, control and high). After the experiment, the levels of transferrin (Tf), transferrin receptor (Tfr), total iron binding capacity (TIBC), hemoglobin (Hb), erythropoietin (EPO), ferritin and hepcidin (HEPC) and liver transcriptome were determined in pigs. DEGs were further analyzed between groups and WGCNA in relation to serum parameters.

I have a number of questions regarding the methods and results of the study:

1) How were iron standards calculated, why do you consider 200 mg/kg a high content?

2) Wujin pig has unique hypoxic adaptation and iron homeostasis. What makes iron homeostasis unique? Is it possible to provide more specific information, preferably in a comparative aspect regarding other breeds?

3) Why did we choose Fold change ≥2 as criteria for screening DEGs?

4) Line 238-241 “The results as shown in Table 3, showed that the feed-to-weight ratio of the low-iron group was higher than that of the high-iron group and the control group. The average daily weight gain of the control group was higher than that of the low-iron and high-iron groups"

Differences between groups are not significant, this statement does not make sense

5) Line 323-324 "3.6. Screening of hypoxic regulation and iron metabolism-related pathways based on WGCNA and pathway enrichment analyses.”

This section is not clear. Figures 8a, 8b, 9 -11 are missing. In which groups WGCNA was performed, regarding what indicators (after all, differences between groups were determined only by the content of HIF-1), what results were obtained.

In general, the purpose of the study is not entirely clear. If Wujin pigs have unique hypoxic adaptation and iron homeostasis, then it is probably better to compare them with other breeds, such as Yorkshire. The choice of iron doses also leaves questions, since in my opinion 200 mg/kg is not a high dose for pigs. Also, how iron doses relate to adaptation. Perhaps these issues need to be addressed in more detail in the introduction.

Regarding the results, it is also somehow not clear; it turns out that ultimately only the signaling pathways of HIF-1 and transforming growth factor-beta (TGF-β) in the regulation of hypoxia and iron metabolism were identified. Meanwhile, liver transcriptome sequencing analysis identified 155 differentially expressed genes (DEGs) between the low-iron and control groups, 229 DEGs between the high-iron and control groups, and 279 DEGs between the low- and high-iron groups. This all somehow remained behind the scenes. It is necessary to present these results at least as additional material.

Lots of questions regarding the WGCNA analysis, perhaps including the necessary figures would provide more information about what was done and what was obtained.

Author Response

Comment 1:[How were iron standards calculated, why do you consider 200 mg/kg a high content?]

Response 1:[The piglet diets in this trial were based on the National Research Council (NRC) Swine Nutrition Requirements (2012) and Swine Nutritional Requirements (GB/T 39235-2020) as a reference, in which the recommended addition of iron, a mineral element, for piglets weighing  8–25 kg is 90 mg/kg, taking into account the health of the piglets. Therefore, we believe that 200 mg/kg for the high iron group is relatively high.]

Comment 2:[Wujin pig has unique hypoxic adaptation and iron homeostasis. What makes iron homeostasis unique? Is it possible to provide more specific information, preferably in a comparative aspect regarding other breeds?]

Response 2: [Thank you for your suggestion. In the article "Hypoxia adaptation in Yunnan Wujin pigs and Youdai Wuhu pigs", we compared the hypoxia adaptation of Wujin pigs with that of other pig breeds and found that the physiological parameters of hypoxia adaptation in Wujin pigs were significantly higher than those in Yuejin Wuhu pigs, which were the progeny of crossbreeding between female Wujin pigs and male Duroc pigs. Therefore, we hypothesized that Wujin pigs have unique adaptive characteristics. The question about what makes iron homeostasis different, which is what we explored here, may require more experiments to verify, and this is something we need to investigate in the future.

Comment 3: [Why did we choose Fold change ≥2 as criteria for screening DEGs?

Response 3: [The selection of screening metrics for DEGs is based on empirical values in general, Fold change ≥2 is based on the data from this experiment, and according to the results of our current experiments, the number of differential genes screened by Fold change ≥2 and P<0.05 is relatively high.

Note 4: [Line 238-241 “The results as shown in Table 3, showed that the feed-to-weight ratio of the low-iron group was higher than that of the high-iron group and the control group. The average daily weight gain of the control group was higher than that of the low-iron and high-iron groups"

Differences between groups are not significant, this statement does not make sense]

Response 4: [Thank you for pointing this out. We have changed the text appropriately and marked them in blue.

Comment 5: [Line 323-324 "3.6. Screening of hypoxic regulation and iron metabolism-related pathways based on WGCNA and pathway enrichment analyses.”

Response 5: [We apologize for the poor description. Relevant content in the text has been corrected and marked in blue.

Comment 6: [This section is not clear. Figures 8a, 8b, 9 -11 are missing. In which groups WGCNA was performed, regarding what indicators (after all, differences between groups were determined only by the content of HIF-1), what results were obtained.

Response 6: [We apologize for the unclear description. Relevant content has been added to the text.

Comment: [In general, the purpose of the study is not entirely clear. If Wujin pigs have unique hypoxic adaptation and iron homeostasis, then it is probably better to compare them with other breeds, such as Yorkshire. The choice of iron doses also leaves questions, since in my opinion 200 mg/kg is not a high dose for pigs. Also, how iron doses relate to adaptation. Perhaps these issues need to be addressed in more detail in the introduction.

RESPONSE: [The aim of this experiment was mainly to investigate the association between iron metabolism and hypoxia acclimatization in the hope of clarifying the role of iron nutrition and its regulatory mechanisms in highland animals. From this article, we mainly showed the characteristics of Wujin pigs, in addition, we also conducted experiments with other breeds (e.g., Duroc pigs), which were not shown in this article due to the large number of words. The selection of iron dosage in this paper followed the National Research Council (NRC) Nutritional Requirements for Swine (2012) and Nutritional Requirements for Swine (GB/T 39235-2020). In the introduction, the text is supplemented with detailed studies related to iron dosage and hypoxia adaptation to the environment.

Comment: [Regarding the results, it is also somehow not clear; it turns out that ultimately only the signaling pathways of HIF-1 and transforming growth factor-beta (TGF-β) in the regulation of hypoxia and iron metabolism were identified. Meanwhile, liver transcriptome sequencing analysis identified 155 differentially expressed genes (DEGs) between the low-iron and control groups, 229 DEGs between the high-iron and control groups, and 279 DEGs between the low- and high-iron groups. This all somehow remained behind the scenes. It is necessary to present these results at least as additional material.

RESPONSE: [The primary focus of this experiment was to determine the association between iron metabolism and hypoxia adaptation. From the results, it can be seen that there are signaling pathways for both iron metabolism and hypoxia detection, i.e., HIF-1 and transforming growth factor-β (TGF-β), which play an important role in the regulation of iron metabolism in Wujin pigs. This is the characteristic of iron metabolism in Wujin pigs. Next, we will compare pig breeds (e.g., Duroc) to determine if Wujin pigs have unique iron regulation. There are other abundant pathways that are not presented in the main manuscript, but we can present them as additional material. In addition, liver transcriptome sequencing analysis yielded differential genes between treatment groups, which we have uploaded as supplemental material.

Comment: [Lots of questions regarding the WGCNA analysis, perhaps including the necessary figures would provide more information about what was done and what was obtained.

RESPONSE: [The results of the WGCNA analyses are presented in more detail in the main text.

Round 2

Reviewer 2 Report

Comments and Suggestions for Authors

Accept in present form